

# Microorganism-regulated autophagy in gastrointestinal cancer

Jun-Yu Xu, Jiao-Xiu Fan, Min Hu and Jun Zeng

Chongqing Normal University, Chongqing, China

## ABSTRACT

Gastrointestinal cancer has always been one of the most urgent problems to be solved, and it has become a major global health issue. Microorganisms in the gastrointestinal tract regulate normal physiological and pathological processes. Accumulating evidence reveals the role of the imbalance in the microbial community during tumorigenesis. Autophagy is an important intracellular homeostatic process, where defective proteins and organelles are degraded and recycled under stress. Autophagy plays a dual role in tumors as both tumor suppressor and tumor promoter. Many studies have shown that autophagy plays an important role in response to microbial infection. Here, we provide an overview on the regulation of the autophagy signaling pathway by microorganisms in gastrointestinal cancer.

## INTRODUCTION

In recent years, the morbidity and mortality of gastrointestinal cancer have been increasing. According to Global Cancer Statistics, there are about 19.29 million new cancer cases and nearly 9.96 million cancer deaths worldwide in 2020 (*Sung et al., 2021*; *Kocarnik et al., 2021*). Approximately 100 trillion microorganisms colonize the human gastrointestinal tract including bacteria, fungi and viruses (*Sender, Fuchs & Milo, 2016*; *Thursby & Juge, 2017*). Accumulating evidence has suggested that imbalance of the colonized microorganisms is associated with gastrointestinal carcinogenesis (*Valdes et al., 2018*; *Kåhrström, Pariente & Weiss, 2016*; *Wroblewski, Peek & Coburn, 2016*; *Wong & Yu, 2019*). *Fusobacterium nucleatum* (*F. nucleatum*), a kind of Gram-negative anaerobic bacterium, increases the risk of colorectal cancer (CRC) (*Ng et al., 2019*). The relationship between autophagy and tumorigenesis has been studied in recent years. Many studies have revealed that autophagy-related molecules are expected to be potential tumor therapeutic targets and biomarkers for tumor prognosis (*Si et al., 2022*; *Tong et al., 2021*). Advances in the relationship among autophagy, microorganisms and tumorigenesis have attracted increasing attention. By clearly elucidating the significance of microbial regulation of autophagy signaling pathway in gastrointestinal cancer in this review, we provide feasible directions and ideas for further research to follow.

## SURVEY METHODOLOGY

In order to ensure that this review is a comprehensive and reasonable presentation of the significance of the study, we mainly obtained the relevant content from the officially

Corresponding author
Jun Zeng, zengjun_2012@163.com

reported data as well as from the extensive literature read. GeenMedical database was used for related literature search using the keyword "gastrointestinal cancer", "Autophagy", "microorganisms", and "tumorigenesis". According to the Global Cancer Statistics 2020 report, the dangers posed by cancers of the gastrointestinal tract have attracted widespread attention and research. In the process of reading the literature, we found that one of the main factors causing the development of gastrointestinal cancers is gut microorganisms, and we noted the close relationship between tumors and autophagy. Therefore, this review aims to elucidate the significance of modulation of the autophagy signaling pathway by microorganism in gastrointestinal cancer.

## AN OVERVIEW OF AUTOPHAGY

Autophagy is a multistep process in which double membrane vesicles encapsulating part of the cytoplasm and damaged organelles are degraded by lysosomes and recycled for cellular metabolic needs and renewal of certain organelles (*Levine & Klionsky, 2004*; *Shintani & Klionsky, 2004*). Till now, the regulation of cancer cell autophagy has become an effective strategy in cancer treatment (*Cordani et al., 2021*; *Salimi-Jeda et al., 2022*).

### Autophagy classification

Autophagy can be classified into macroautophagy, microautophagy and chaperone-mediated autophagy (CMA) based on the pathway by which cellular contents are transported into the lysosome (Fig. 1) (*Jacob et al., 2017*). During the process of macroautophagy, cytoplasmic contents or organelles to be degraded are wrapped by double-membraned autophagosomes from the endoplasmic reticulum (ER) (*Axe et al., 2008*) and Golgi apparatus (*Yen et al., 2010*) which will fuse with lysosomes to form autolysosomes.

The contents will be degraded into small biomolecules and released into the cytoplasm, even outside the cell for recycling (*Mizushima et al., 2001*). Microautophagy is the process by which the lysosomal membrane invaginates and directly wraps the damaged organelles, then transports the cargos to the lysosomes (*Gorrell et al., 2021*). Finally, CMA is a selective process in which unfolded proteins recognize and bind to the molecular chaperone and enter the lumen of the lysosomes directly for degradation (*Kaushik, Kiffin & Cuervo, 2007*). Macroautophagy, usually referred to simply as autophagy, is the subject of this review.

### The molecular mechanisms of autophagy

The whole process of autophagy includes six key steps: initiation, nucleation, prolongation, maturation, fusion, and degradation (*Levy, Towers & Thorburn, 2017*). The following signal molecules are involved in autophagy: the ULK1 complex (ULK1-Atg13-FIP200), the type III phosphatidylinositol 3-kinase (PI3K) complex (Class III PI3K (VPS34)-Beclin-1-Atg14), the Atg12-Atg5-Atg16 ubiquitin complex and LC3-II-PE ubiquitin complex, *etc*.

The ULK1 complex is involved in autophagy induction (*Hosokawa et al., 2009*). mTOR phosphorylates Atg13, resulting in a low ULK1 activity in normal situation. When cells are starved or hypoxic, the mTOR activity is inhibited, leading to dephosphorylation of Atg13, which will activate ULK1 complex. The activated ULK1 complex will be further transferred

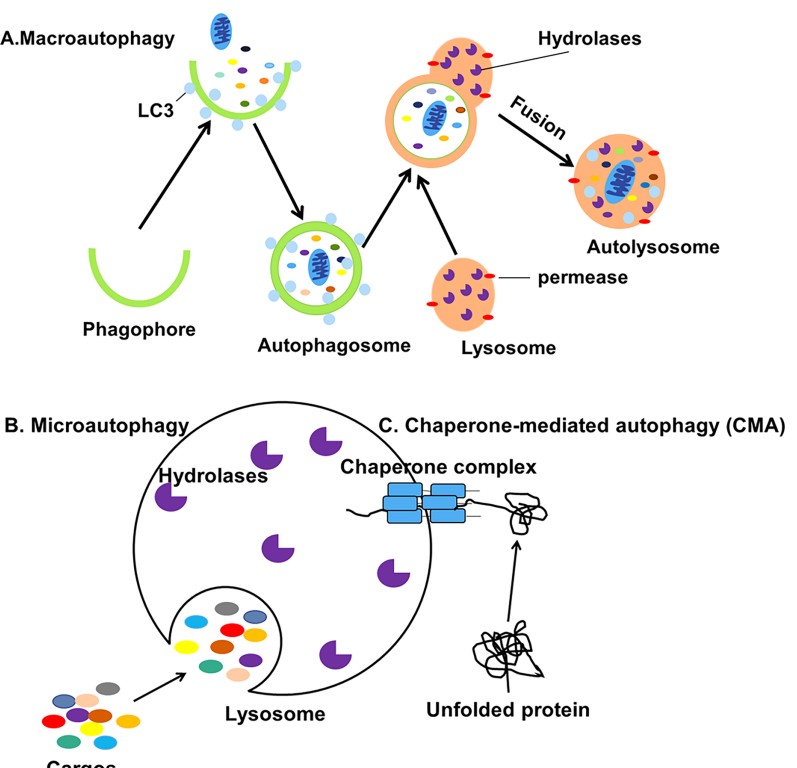

**Figure 1 Three different types of autophagy.** (A) Macroautophagy is the process in which intracellular cargos are wrapped in a bilayer membrane, forming a bilayer structure that forms autophagosomes and fuses with lysosomes. (B) Microautophagy is the pathway by which the lysosomal membrane itself invaginates and directly wraps the cytoplasmic contents. (C) CMA is a selective process in which unfolded proteins recognize and bind to the chaperone complexes in the cytoplasm and enter the lumen of the lysosomes directly for degradation.

from the cavity to the endoplasmic reticulum to induce the membrane formation of autophagosomes (*Pattingre et al., 2008*). AMPK is a positive regulator of autophagy, which can directly inhibit mTOR activity and induce autophagy (*Ge et al., 2021*). In addition, AMPK can directly bind to ULK1 complex and phosphorylate it, thus promoting the progression of autophagy membrane (*Kim et al., 2011*).

The Type III PI3K-Beclin-1-Atg14 complex is involved in the nucleation of autophagosomes (*Kang et al., 2011*). After being activated by the ULK1 complex, it locates to the ER and produces PI3P, mediating the formation of autophagy vesicles (*Levine, Mizushima & Virgin, 2011*). Beclin1 is a key factor in the formation of autophagy, which can bind to anti-apoptosis-related proteins such as Bcl-2, thus playing an important role in regulating autophagy and apoptosis (*Kang et al., 2011*).

The Atg12-Atg5-Atg16 ubiquitin complex is involved in the prolongation of autophagosomes. The formation of the complex requires the participation of ubiquitin activating enzymes E1 and E2. Atg12 is first activated by Atg7 (E1-like enzyme), and then transported to Atg5 through Atg10 (E2-like enzyme) to form a multi-body complex with Atg16, which participates in the extension of autophagosomes (*Tanida, 2011*; *Suzuki et al., 2005*).

LC3II-PE ubiquitin complex plays an essential role in the early stage of autophagy. LC3 can be cut into soluble LC3I by Atg4, and then combined with phosphatidylethanolamine (PE) under the action of Atg7 and Atg3 to form LC3II-PE, which participates in the prolongation of autophagosomes (*Kabeya et al., 2000*). LC3II distributes symmetrically on the inner and outer membrane of autophagosomes. When autophagosomes fuse with lysosomes, LC3II is degraded by hydrolases in lysosomes. The content of LC3II or the ratio of LC3II to LC3I could be an indicator of autophagy (*Kimura et al., 2009*). Moreover, LC3II-PE can transfer the ubiquitinated products to be degraded into the autophagolysosomes through the junction of p62 (*Matsumoto et al., 2011*). P62 degradation is another important indicator of autophagy (*Mizushima, Yoshimori & Levine, 2010*).

And in the process of fusion and degradation, autophagosomes act mainly by forming the autophagolysosomal system with lysosomes (*Kumar et al., 2021*). In the autophagolysosomal system, Transcriptional factor EB (TFEB) plays an important role in the regulation of the expression of multiple genes, including autophagolysosomal components. The nuclear localization of TFEB is regulated by the phosphorylation of extracellular signal-regulated kinase 2 (ERK2), and its activity is modulated by the levels of extracellular nutrients. Interestingly, reactive oxygen species (ROS) play key roles in the autophagolysosomal system and may be critical for synergistic therapeutic interventions (*Kumar et al., 2021*; *Settembre et al., 2011*).

## THE ROLE OF AUTOPHAGY IN TUMORIGENESIS

Autophagy is a form of programmed cell death and plays an important role in maintaining intracellular homeostasis. Autophagy contributes to immunity, infection, cytotoxicity, drug resistance, and tumorignesis (*Dikic & Elazar, 2018*; *Li et al., 2021*; *Mele et al., 2020*).

### Autophagy as a tumor promoter

Autophagy is thought to function as a promoter of tumor progression and is associated with drug resistance in several types of cancer (*Maes et al., 2013*). However, some chemotherapeutic drugs can induce protective autophagy, thereby antagonizing drug-induced apoptosis in cancer cells (Table 1). A recent study has shown that oxaliplatin-induced protective autophagy could partially antagonize apoptosis in gastric cancer MGC803 cells, promoting tumor progression (*Xu et al., 2011*). The expression levels of LC3II have been reported to be positively correlated with the clinical stages in oral squamous carcinoma (OSCC) (*de Lima et al., 2017*). Some normal cells contribute to tumor cell growth by generating nutritional autophagy at the early stage of tumor development (*Maes et al., 2013*; *Katheder et al., 2017*). ATG16L1, an essential signal molecule for autophagy, is expressed in malignant oral cancer cells but not in normal cells, suggesting elevated levels of autophagy in tumors (*Nomura et al., 2009*). Thus, autophagy is likely to be a protective factor for tumor cells, allowing them to survive under stress. A recent study has demonstrated that knockdown of FIP200, a protein involved in autophagy initiation, prevented breast cancer progression, suggesting a role for autophagy in tumorigenesis (*Wei et al., 2011*).

**Table 1 The roles of autophagy in tumorigenesis.**

| Effect on tumorigenesis | Regulatory mechanism | Type of cancer studied | References |
|---|---|---|---|
| Promoter | Akt/mTOR, AQP3 | Gastric cancer | *Xu et al. (2011)*, *Dong et al. (2016)* |
| Promoter | AMPK/HIF-1/ATG16L1, LC3/p62/SQSTM1 | OSCC | *de Lima et al. (2017)*, *Nomura et al. (2009)*, *Lai et al. (2018)*, *Terabe et al. (2018)* |
| Promoter | PERK/eIF2α/ATF4 | Lymphoma | *Hart et al. (2012)* |
| Promoter | Ki-67 index | Gastrointestinal cancer | *Yoshioka et al. (2008)* |
| Promoter | Hypoxia, H-Ras, ROS/DNA damage | Pancreatic cancer | *Fujii et al. (2008)*, *Guo et al. (2011)*, *Yang et al. (2011)* |
| Promoter | FIP200/p62/SQSTM1 | Mammary cancer | *Wei et al. (2011)* |
| Promoter | Atg7/Nrf2, ATG7/K-Ras/P53 | Lung cancer | *Strohecker et al. (2013)*, *Guo et al. (2013)*, *Karsli-Uzunbas et al. (2014)* |
| Promoter | ATG7/AMPK/P53 | Colorectal cancer | *Lévy et al. (2015)* |
| Promoter | ATG7/ER stress | Prostate cancer | *Santanam et al. (2016)* |
| Promoter | K-Ras, HIF-1α/AMPK | Glioblastoma | *Gammoh et al. (2016)*, *Hu et al. (2012)* |
| Promoter | BrafV600E/PTEN/ATG7 | Melanoma | *Xie et al. (2015)* |
| Promoter | K-Ras | Bladder cancer | *Guo et al. (2013)* |
| Suppressor | MAPK/mTOR/p70S6K/Ak, miR-30/Beclin-1 | Gastric cancer | *Zhang et al. (2020)*, *Qian & Yang (2016)*, *Yang & Pan (2015)* |
| Suppressor | CHOP/ROS/ER stress | Melanoma | *Fang et al. (2021)* |
| Suppressor | ERK1/2 signal pathway | Glioblastoma | *Qu et al. (2020)* |
| Suppressor | BECN1, DEDD/Vps34, EBP50/Beclin-1 | Breast cancer | *Aita et al. (1999)*, *Lv et al. (2012)*, *Liu et al. (2015)* |
| Suppressor | PTEN/PI3K/PKB, p53 signal pathway | Colorectal cancer | *Arico et al. (2001)*, *Tasdemir et al. (2008)* |
| Suppressor | mTOR signal pathway | Lymphoma | *Kittipongdaja et al. (2015)* |

## Autophagy as a tumor suppressor

Autophagy can protect cells from cancerization by degrading dysfunctional proteins and organelles and preventing the toxic accumulation (*Liang & Jung, 2010*). Abnormal expression of autophagy-associated genes may lead to pathological changes (*Maiuri et al., 2009*; *Tsuchihara, Fujii & Esumi, 2009*). Some autophagy-associated genes are frequently mutated or absent in many human cancers (*Ionov et al., 2004*; *Aita et al., 1999*). The down-regulation of beclin1 expression has been observed in human breast, ovarian and prostate cancers (*Aita et al., 1999*). Similarly, knockdown of the *atg5* gene and/or the *beclin1* gene in normal cell line (*Karantza-Wadsworth et al., 2007*) has been shown to induce cell transformation.

Some drugs can induce autophagic death in cancer cells (Table 1). A study has showed that berberine, an alkaloid isolated from the Chinese herbal medicine *Coptis chinensis*, may induce autophagy through inhibition of MAPK/mTOR/p70S6K and Akt signaling pathways, thereby suppressing the growth of human gastric cancer cells *in vivo* and *in vitro* (*Zhang et al., 2020*). Berberine may also induce apoptosis in human malignant melanoma cells through activation of the ER stress-mediated autophagy (*Fang et al., 2021*). Berberine

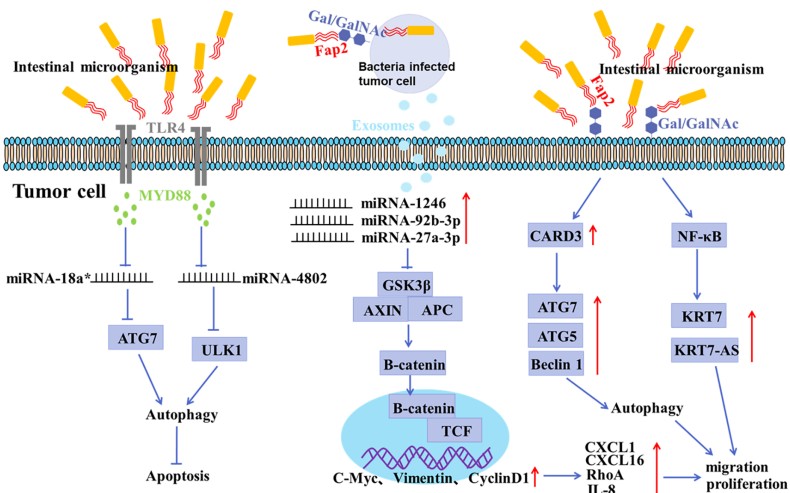

**Figure 2 Mechanisms of gastrointestinal microorganism involved in tumorigenesis.** *F. nucleatum* invades tumor cells through the binding of Fap2 to Gal/GalNAc expressed by tumor cells, induces the secretion of IL-8 and CXCL1, and promotes the metastasis of tumor cells; *F. nucleatum* acts on tumor cells *via* TLR4 and MYD88, resulting in selective deletion of miR-18a* and miR-4802 expression, which in turn leads to autophagy activation and thus promotes chemoresistance in cancer patients; *F. nucleatum* stimulates tumor cells to produce miR-1246/92b-3p/27a-3p and CXCL16/RhoA/IL-8 enriched exosomes, which are delivered to uninfected cells to promote metastatic behaviors; *F. nucleatum* promotes metastasis in CRC through upregulation of CARD3 and activation of autophagic signaling by ATG7/ATG5/Beclin 1; *F. nucleatum* promotes tumor cell metastasis by NF-κB/KRT7/KRT7-AS pathway.

may also induce autophagic death in acute lymphoblastic leukemia through inactivation of AKT/mTORC1 signaling (*Liu et al., 2020*). In addition, berberine may induce autophagy in glioblastoma through the ERK1/2 signaling pathway, thereby increasing sensitivity to chemotherapeutic drugs (*Qu et al., 2020*). A study has showed that artesunate, a kind of antimalarial drug, can act as an autophagy inducer to suppress colorectal cancer progression in a ROS-dependent manner (*Huang et al., 2022*). Aloe gel glucomannan can induce colon cancer cell death *via* the PINK1/Parkin mitochondrial autophagy pathway (*Zhang et al., 2022*). Moreover, inappropriate degradation of components during the autophagy process may bring cytotoxicity, ultimately leading to autophagic cell death (*Qian & Yang, 2016*). In conclusion, autophagy is a double-edged sword for tumor cells.

# MICROORGANISMS ASSOCIATED WITH TUMORIGENESIS

## The intestinal microorganisms and colorectal cancer

In recent years, the association between microorganisms and cancer development has been studied. Approximately 100 trillion bacteria colonize the human intestine (*Yu et al., 2015*). Intestinal microbes interact with the human body in long-term coevolution, which are closely associated with some physiological and pathological activities, such as obesity, diabetes, cardiovascular disease, *etc*. (*Neish, 2009*; *Schulz et al., 2014*; *Wang et al., 2021b*; *De Vadder et al., 2014*; *Poli, 2020*). Accumulating evidence has shown that imbalance of intestinal microbiota is closely related to tumorigenesis, as shown in Fig. 2 (*Neish, 2009*; *Sonnenberg & Artis, 2012*). Targeting the intestinal microorganisms may be a potent

strategy in cancer treatment (*Fong, Li & Yu, 2020*; *Song et al., 2020*; *Ji et al., 2020*; *Kaźmierczak-Siedlecka et al., 2020*; *Kim et al., 2020*; *Inamura, 2021*; *Baffy, 2020*; *Cueva et al., 2020*; *Peterson, Bradley & Ronai, 2020*; *Tao et al., 2020*).

There is growing evidence of a direct link between intestinal microbiota imbalance and colorectal cancer (CRC) (*Liu et al., 2021*; *Raskov, Burcharth & Pommergaard, 2017*). It has been shown that the abundance of *F. nucleatum* positively correlated with CRC in clinicopathological stages (*Castellarin et al., 2012*; *Kostic et al., 2012*). The mucosal microbiota in normal tissues, adenomatous polyps, and adenocarcinoma tissues were compared, and the results showed that CRC tissues at early stage had a significant increase in the abundance of *Fusobacterium*, *Parvimonas*, *Gemella*, and *Leptotrichia*, and a decrease in the abundance of *Bacteroides*, *Blautia*, and *F. prausnitzii*, indicating an oncogenic role of microbiota imbalance (*Nakatsu et al., 2015*). In addition, the abundance of *Peptostreptococcus*, *Parvimonas* and *Fusobacterium* in *CRC tissues* was significantly different from that in paracancerous mucosa tissues (*Nakatsu et al., 2015*; *Warren et al., 2013*). It has been shown that *F. nucleatum* may invade tumor cells by binding Fap2 to Gal/Gal NAc expressed by tumor cells, mediating multidrug resistance of tumor cells (*Abed et al., 2016*). *F. nucleatum* can promote the metastasis of tumor cells by inducing the secretion of IL-8 and CXCL1 (*Casasanta et al., 2020*). Moreover, Exosomes derived from *F. nucleatum*-infected CRC cells may facilitate non-infected tumor cell metastasis by selectively carrying miR-1246/92b-3p/27a-3p and CXCL16 (*Guo et al., 2020*). Recent studies have confirmed that the anaerobic bacterium *peptostreptococcus anaerobius* can also promote the development of CRC (*Tsoi et al., 2017*). *Enterotoxigenic Bacteroides fragilis* can promote malignant behaviors by inhibiting miR-149-3PF packaged *in vitro* (*Cao et al., 2021*).

## The gastric microorganisms and gastric cancer

As the third leading cause of cancer-related deaths worldwide, gastric cancer and its risk factors and prevention have been extensively studied (*Noto & Peek, 2017*). Gastric bacterial communities have been shown to be associated with gastric malignancy. *H. pylori*, a Gram-negative bacterium that colonizes the gastric epithelium, which is classified as a class I carcinogen by the World Health Organization (*Moodley et al., 2012*; *De Meyer et al., 2015*). *H. pylori* infection is thought to be the main cause of gastric cancer, but its exact mechanisms have not been fully understood (Fig. 3). *H. pylori* releases various virulence factors, such as vacuolating cytotoxin A (VacA) and the effector protein cytotoxin-associated gene A (CagA), to promote the development of gastric cancer (*Ferreira, Machado & Figueiredo, 2014*). Prolonged infection with *H. pylori* can lead to gastric atrophy, resulting in hyperacidity or decreased gastric acid production. Notably, *H. pylori* infection and the following change in the acidity of the gastric environment may further lead to alterations in the gastric microbiota (*Espinoza et al., 2018*). In addition, *H. pylori* infection was reported to increase the expression of VCAM1 in cancer-associated fibroblasts (CAFs) *via* JAK/STAT1 signaling pathway in gastric carcinoma, and the level of VCAM1 in patients with gastric cancer was positively correlated with tumor progression and a poor prognosis. Moreover, the interaction between CAF-derived VCAM1 and

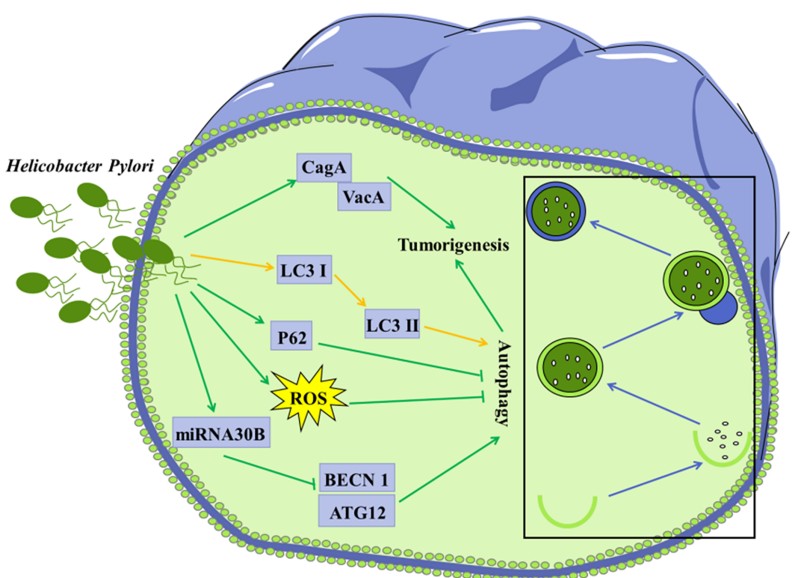

**Figure 3 Autophagy is considered to be involved in certain microorganism-mediated tumorigenesis.** Mechanisms of *H. Pylori*-regulated autophagy in gastric cancer cells *H. pylori* releases VacA and effector protein CagA, which promote the progression of gastric cancer; Infection of *H. pylori* induces conversion of LC3 I to LC3 II in tumor cells; Invasion of *H. pylori* into tumor cells leads to elevated ROS and p62 levels, further inhibiting autophagy; *H. pylori* represses the expression of BECN1 and ATG12 through up-regulation of miRNA30B, thereby inhibiting autophagy.

integrin αvβ1/5 could promote gastric cancer cell invasion both *in vitro* and *in vivo* (*Shen et al., 2020*). A recent study showed that *Firmicutes* and *Actinobacteria* were over-distributed in gastric cancer compared to chronic gastritis (*Ferreira et al., 2018*). These findings suggest that microbial imbalance increases the risk of gastric cancer. Till now, the functional role of microbial communities in gastric tumorigenesis and its pathogenic mechanisms have not been well understood.

# MODULATION OF THE AUTOPHAGY BY THE MICROORGANISMS COLONIZED IN GASTROINTESTINAL TRACT

It is becoming increasingly clear that imbalance of the microorganisms in the gastrointestinal tract contributes to the development of gastric cancer and CRC. Autophagy is considered to be involved in certain microorganism-mediated tumorigenesis (Figs. 2 and 3) (*Castrejón-Jiménez et al., 2015*; *Wang et al., 2021a*).

Accumulating evidence has shown that invasion *of H. pylori* can interfere with autophagy in gastric epithelial cells (*Deen et al., 2013*; *Shao et al., 2022*). Autophagy in cells infected with *H. pylori* may be a way to clear invaded *H. pylori*, thus protecting other gastric epithelial cells against infection with *H. pylori*. It was shown that conversion of LC3 I to LC3 II in the *H. pylori*-infected gastric epithelial cells represented a host protective mechanism to limit *H. pylori*-induced cellular damage (*Terebiznik et al., 2009*). *In vivo* and *in vitro* studies have revealed that *H. pylori* could encroach on the autophagy pathway in

gastric mucosa cells, leading to elevated levels of ROS, which contribute to gastric tumorigenesis (*Raju et al., 2012*). *H. pylori* was reported to down-regulate the expression of autophagy-associated proteins BECN1 and ATG12, leading to tumorigenesis (*Tang et al., 2012*).

Other microorganisms have also been involved in autophagy-mediated tumor initiation and progression. *F. nucleatum* in patients' tissues has been reported to be strongly associated with recurrence and survival rates in CRC patients (*Yu et al., 2017*). A study found that *F. nucleatum* might act on CRC cells through TLR4 and MYD88, leading to selective deletion of miR-18a* and miR-4802 expression, which in turn led to autophagy activation and thus promoted chemotherapy resistance in CRC patients (*Yu et al., 2017*). *F. nucleatum* has also been shown to promote metastasis by activating autophagy signal pathways in CRC (*Chen et al., 2020a*). *F. nucleatum* also promotes metastasis in colorectal cancer by upregulating KRT7/KRT7-AS through activation of the NF-κB pathway (*Chen et al., 2020b*).

## EFFECT OF DIET ON MICROORGANISM-MEDIATED TUMORIGENESIS

While genetic factors play a large role in cancer risk, as many as 50% of cancers may be preventable through a variety of lifestyle changes (*Klein, 2021*). Although cancer is a multifactorial disease, diet is one of the largest sources of modifiable risk. It was estimated that diet accounted for 30–35% of the total risk associated with carcinogenesis, and this percentage might be greater for some categories of cancer, such as colorectal cancer (*O'Keefe et al., 2015*). Notably, gastrointestinal microorganisms could alter dietary substrates, leading to the production of microbial metabolites such as short chain fatty acids (SCFA), which were important in induction of apoptosis in cancer cells, regulation of tumor suppressor gene expression through inhibition of histone deacetylases, and regulation of cellular glucose metabolism (*O'Keefe et al., 2015*). Some data indicated that gastrointestinal microorganisms could directly regulate the metabolism of some chemotherapeutic drugs and the activity of host enzymes (*van Duynhoven et al., 2011*). Therefore, a better understanding of the dynamic interactions between gastrointestinal microbes, diet, and cancer risk is essential to guide future cancer prevention and treatment.

## CONCLUSIONS AND PROSPECTS

The gastrointestinal microbiota plays an important role in maintaining normal physiological processes in the human body. Abnormalities in the microbiota may eventually lead to various diseases including obesity, diabetes, cardiovascular disease and even cancer. A causal relationship between microbiota and gastrointestinal tumors has been gradually revealed. In fact, several studies have shown that target on the microbiota, especially specific bacteria, may be potential strategies for the prevention, diagnosis and treatment (*Chen et al., 2021*).

Autophagy is considered to be a self-protective way for cells under stress, which can not only promote the development of tumors, called protective autophagy, but also inhibit the progress of tumors, called cytotoxic autophagy. Interestingly, some chemotherapeutic

drugs can induce apoptosis of cancer cells while inducing protective autophagy (*Zhou et al., 2019*), which awaits further investigation. Overall, the role of autophagy in tumorigenesis varies depending on the stages and types of tumors (*Wang et al., 2021a*). Although the exact molecular mechanisms of autophagy in gastrointestinal tumors are unknown, there is no doubt that autophagy is closely related to tumor initiation, progression, prognosis, and treatment.

Accumulating evidence has shown that microbes in the gastrointestinal tract may be involved in tumorigenesis in an autophagy-dependent way (*Raju et al., 2012*). The association between microbe-regulated autophagy and gastrointestinal tumors is complex. Autophagy can protect epithelial cells against infection with microbes colonized the gastrointestinal tract. It can also act as an accomplice of intestinal microorganisms, gradually contributing to inflammation, even tumorigenesis. Interestingly, diets and lifestyles are closely related to intestinal microbiota. High-fat intake is significantly correlated with the incidence of CRC (*Keum & Giovannucci, 2019*). Intake of less pickled food is considered an important way to prevent gastric cancer (*Ren et al., 2012*). In all, microorganism-regulated autophagy may contribute to new insights into the occurrence, prevention and treatment of gastrointestinal cancers.

## LIST OF ABBREVIATIONS

| | |
|---|---|
| **H. pylori** | *Helicobacter pylori* |
| **F. nucleatum** | *Fusobacterium nucleatum* |
| **CRC** | colorectal cancer |
| **CMA** | chaperone-mediated autophagy |
| **ER** | endoplasmic reticulum |
| **PI3IK** | phosphatidylinositol 3-kinase |
| **PE** | phosphatidyl ethanolamine |
| **TFEB** | Transcriptional factor EB |
| **ERK2** | extracellular signal-regulated kinase 2 |
| **ROS** | reactive oxygen species |
| **OSCC** | oral squamous carcinoma |
| **VacA** | vacuolating cytotoxin A |
| **CagA** | cytotoxin-associated gene A |
| **CAFs** | cancer-associated fibroblasts |
| **SCAF** | short chain fatty acids |

### Funding

This work was supported by the grants from the National Natural Science Foundation of China (81502131), the Natural Science Foundation of Chongqing (cstc2018jcyjAX0573, cstc2018jcyjAX0816) and the Scientific and Technological Research Program of Chongqing Municipal Education Commission (KJ202000541975044). The funders had no

role in study design, data collection and analysis, decision to publish, or preparation of the manuscript.

## Grant Disclosures

The following grant information was disclosed by the authors:

National Natural Science Foundation of China: 81502131.

Natural Science Foundation of Chongqing: cstc2018jcyjAX0573, cstc2018jcyjAX0816.

Scientific and Technological Research Program of Chongqing Municipal Education Commission: KJ202000541975044.

## Competing Interests

The authors declare that they have no competing interests.

## Author Contributions

- Jun-Yu Xu conceived and designed the experiments, performed the experiments, prepared figures and/or tables, authored or reviewed drafts of the article, and approved the final draft.
- Jiao-Xiu Fan conceived and designed the experiments, analyzed the data, prepared figures and/or tables, and approved the final draft.
- Min Hu conceived and designed the experiments, performed the experiments, analyzed the data, prepared figures and/or tables, and approved the final draft.
- Jun Zeng conceived and designed the experiments, authored or reviewed drafts of the article, and approved the final draft.

## Data Availability

This is a literature review and did not utilize raw data.

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
