# Peer review of "Microorganism-regulated autophagy in gastrointestinal cancer"

_PeerJ, doi:10.7717/peerj.16130_

## Round 0.1 · original submission · Major Revisions

Based on review reports received from reviewers, authors need to revise the current manuscript thoroughly as per the raised concerns.

Reviewer 1 ·

Basic reporting

English language editing is strongly recommended. The authors may seek help of professional English writers.
Additionally, some of the full forms are not defined on their first usage; please do so..

Experimental design

No comment.

Validity of the findings

No comments.

·

Basic reporting

No comment

Experimental design

No comment

Validity of the findings

No comment

Additional comments

The authors need to be congratulated on taking up a crucial upcoming topic, guiding future research, and doing an extensive review. Have a few comments:
Line 30/31: Authors may make it more impactful by quoting data like ‘Gastric cancer is the --- most common cancer in the world with the --- common cause of cancer-related mortality’
Line 34: English language needs improvement
Line 85: Do the authors mean ‘degradation’ Please check for typo.
Line 215: Need reference for the statement

If English is not the native language of the authors, then they may get the manuscript edited by an editing service to improve the manuscript

Reviewer 3 ·

Basic reporting

no comment

Experimental design

1. The microorganisms mentioned in this review are primarily pathogenic bacteria. It would be better to include commensal microorganisms as well in order to highlight the diverse effects the microorganisms have on autophagy regulation in gastrointestinal cancer.

2. It would be beneficial to provide more specific examples and case studies that illustrate the impact of microorganism dysregulation on gastrointestinal cancer treatment responses. It was only briefly mentioned in part 5 Effect of diet on microorganism-mediated tumorigenesis.

Validity of the findings

This review needs more discussions about the limitations and challenges faced in studying microorganism-regulated autophagy in gastrointestinal cancer. The authors need to address the need for further research, highlight knowledge gaps, and suggest future investigation directions.

Additional comments

This review paper offers a comprehensive overview of the intricate interplay between microorganisms and autophagy in gastrointestinal cancer. The authors have effectively synthesized the existing literature, providing valuable insights into the mechanisms by which microorganisms influence autophagy and its implications for gastrointestinal tumorigenesis.

One of the strengths of this review is the inclusion of graphical representations or schematic diagrams depicting the molecular mechanisms involved in microorganism-mediated autophagy regulation. The authors have skillfully discussed the molecular mechanisms through which microorganisms manipulate autophagy, including the modulation of signaling pathways and host immune responses. This comprehensive approach contributes to a better understanding of the complex relationship between microorganisms and gastrointestinal cancer.

In conclusion, this review paper presents a comprehensive and well-organized analysis of the interplay between microorganisms and autophagy in gastrointestinal cancer. With the suggested improvements, this review has the potential to serve as a valuable resource for researchers and clinicians interested in understanding the intricate connections between microorganisms, autophagy, and gastrointestinal tumorigenesis.

·

Basic reporting

The manuscript is professionally written. It is clear and easy to follow. Except for a minor typographical error (Line 162 – “correlated with CRC (in) clinicopathological stages” missing the word “in”), the manuscript does not have any grammatical errors.
The review discusses a broad interdisciplinary topic which is relevant in light of different ongoing scientific studies. Enough background is discussed to introduce the topic and discuss the ongoing research, except for a few key publications which are missed by the author. (Raskov, H., Burcharth, J., & Pommergaard, H. C. (2017). Linking gut microbiota to colorectal cancer. Journal of cancer, 8(17), 3378.; Wang, Y., Du, J., Wu, X., Abdelrehem, A., Ren, Y., Liu, C., ... & Wang, S. (2021). Crosstalk between autophagy and microbiota in cancer progression. Molecular Cancer, 20(1), 1-19.; Shao, B. Z., Chai, N. L., Yao, Y., Li, J. P., Law, H. K. W., & Linghu, E. Q. (2022). Autophagy in gastrointestinal cancers. Frontiers in Oncology, 4596.) These publications are highly relevant to the topic being discussed.
The publication Shao, B. Z., Chai, N. L., Yao, Y., Li, J. P., Law, H. K. W., & Linghu, E. Q. (2022). Autophagy in gastrointestinal cancers. Frontiers in Oncology, 4596. is a recent review done in this field. Not enough evidence is shown to justify the current review in light of this previous publication.

Experimental design

The survey methodology is well elucidated and it included a comprehensive description of the subject and what the aim of this manuscript is. It will however, benefit from the inclusion of the current review mentioned in the previous section (Shao, B. Z., Chai, N. L., Yao, Y., Li, J. P., Law, H. K. W., & Linghu, E. Q. (2022). Autophagy in gastrointestinal cancers. Frontiers in Oncology, 4596.) and how this review will differs in terms of point of view or recent updates in the field.
Except for the publications mentioned, the sources are well cited. There are a few instances where the source material doesn't match the inference drawn in the manuscript. They are highlighted below.
“According to Global Cancer Statistics, there are about 3.0 million new cancer cases and nearly 1.7 million cancer deaths 43 worldwide in 2020 [1, 2]” – Statistics do not match with reference paper #1.
“It has been reported (line 46) that infection with Helicobacter pylori (H. pylori) is positively correlated with the incidence of gastric cancer [9]” – no mention of H. pylori in reference paper #9.
The manuscript is well organized with sections introducing the topics and how they correlate with each other. The abbreviation section and the images are helpful in understanding the topic.
Table 1 summarizes the differing role of autophagy in tumorigenesis. It will make sense to refer to this table in the manuscript.

Validity of the findings

The argument and the goals of the review as defined in the abstract and introduction is met with sufficient evidence in the main body of the manuscript and the conclusion is intuitively drawn from the evidence presented in the manuscript.
However, given the recent review on a similar topic, it is unclear how this review will further the understanding of the field already covered in some of the aforementioned reviews. The main goal of a review is to shed light on a novel topic or interrogate an existing one from a novel angle which will open new avenues for researches to focus on. Given the evidence that has been laid out in the previous section, it is unclear how the authors have addressed this in this manuscript.

---

## Round 0.2 · accepted · Accept

Based on the reviewers' reports the current manuscript is accepted.

Reviewer 1 ·

Basic reporting

Dear authors,
I thank you for submitting the revised manuscript titled ‘’Microorganism-regulated autophagy in gastrointestinal cancer”. The paper explores the complex interplay between gut microbiota, autophagy and gastrointestinal cancer. The methodology followed is systematic and meticulously explains the developments in the area. I believe the paper will attract wide readership. I congratulate the team for their dedication and contribution to the field.
Best!
Uzma

Experimental design

The study design is systematic and comprehensively explains the developments in this area. The reference sources are adequately cited.

Validity of the findings

The paper may be a valuable and updated contribution in the field. The arguments laid out in the introduction are reasonably addressed.

·

Basic reporting

Modified as suggested by reviewers

Experimental design

Modified as suggested by reviewers

Validity of the findings

Modified as suggested by reviewers

Additional comments

The authors have made all essential changes as advised.

Reviewer 3 ·

Basic reporting

It is not necessary to list all the authors for the references.

All other concerns have been addressed for reviewer 3.

Experimental design

no comment

Validity of the findings

no comment